# Engineering the Exchange Spin Waves in Graded Thin Ferromagnetic Films

**DOI:** 10.3390/nano12244361

**Published:** 2022-12-07

**Authors:** Igor Yanilkin, Amir Gumarov, Igor Golovchanskiy, Bulat Gabbasov, Roman Yusupov, Lenar Tagirov

**Affiliations:** 1Institute of Physics, Kazan Federal University, 420008 Kazan, Russia; 2FRC Kazan Scientific Centre of RAS, Zavoisky Physical-Technical Institute, 420029 Kazan, Russia; 3National University of Science and Technology MISiS, 119049 Moscow, Russia; 4Advanced Mesoscience and Nanotechnology Centre, Moscow Institute of Physics and Technology, 141700 Dolgoprudny, Russia

**Keywords:** PdFe alloy, graded magnetic materials, standing spin waves, magnetic resonance

## Abstract

The results of experimental and theoretical studies of standing spin waves in a series of epitaxial films of the ferromagnetic Pd_1−*x*_Fe*_x_* alloy (0.02 < *x* < 0.11) with different distributions of the magnetic properties across the thickness are presented. Films with linear and stepwise, as well as more complex Lorentzian, sine and cosine profiles of iron concentration in the alloy, and thicknesses from 20 to 400 nm are considered. A crucial influence of the magnetic properties profile on the spectrum of spin wave resonances is demonstrated. A capability of engineering the standing spin waves in graded ferromagnetic films for applications in magnonics is discussed.

## 1. Introduction

Spin waves have been known for more than 90 years, but recently, the interest in them has increased dramatically due to the rapid development of magnonics, which, unlike electronics, for example, studies the transfer of energy and information not by the electric current, but rather by the current of magnons. Moreover, magnonics has evolved to a large field in engineering that develops devices based on wave principles, such as magnon transistors, switches, sensors and inductors, as well as media for magnetic information recording (see reviews [1,2,3,4] and references therein). Existing technologies that allow the fabrication of nanostructures with a variety of physical and geometric parameters, as well as new experimental methods for studying the dynamics of high-frequency magnetization, open up broad possibilities for the engineering of magnetic devices that can be used to effectively manipulate magnetization waves, their excitation and propagation [2,5,6,7].

Nowadays, spin waves are studied intensely in thin ferromagnetic films and heterostructures [8,9,10,11]. In the majority of the reports, homogeneous films or magnonic crystals with a homogeneous periodic structure are investigated [12,13]. However, the studies of the spin waves in films with artificially created inhomogeneity of magnetic properties across the thickness, known in the literature as graded magnetic materials, have been performed mostly in theory [14,15,16,17,18]. These papers report that by varying the profile of the magnetic properties distribution over the thickness, it is possible to manipulate the spectrum of spin waves in thin magnetic films [14].

Our recent studies have shown that Pd–Fe alloys are promising and convenient model materials that allow the synthesis of high-quality thin epitaxial films with a predetermined distribution of magnetic inhomogeneity across the thickness [19]. This study aimed at the experimental investigation of the possibility to engineer the spin wave spectra by means of controlling a magnetic property profile in graded films of Pd–Fe alloy.

## 2. Materials and Methods

### 2.1. Description of Samples and Research Methods

The synthesized collection of epitaxial Pd–Fe alloy films with variable distribution profiles of a composition across the thickness included (see Table 1):Films with a fixed iron concentration gradient from 2 at.% to 10 at.%—linear, with thicknesses in the range of 50–400 nm;Bilayer and trilayer structures with different iron concentrations in each layer and a total thickness in the range of 20–220 nm. Each individual layer was epitaxial and uniform in composition. The layers were deposited on top of each other in a single run. Several concentrations of iron were chosen in the bilayer and trilayer samples: 4 at.% and 8 at.% in the bilayer and 2 at.% and 10 at.% in the trilayer ones;Films with non-linear composition distributions across the thickness—Lorentzian, sine and cosine, with thicknesses in the range of 200–400 nm.

All Pd–Fe alloy films investigated in this work were epitaxial, continuous and single-crystalline. The high-purity (99.95%) Pd and Fe metals were evaporated from the effusion cells from Createc Fischer & Co GmbH (Erligheim, Germany) and deposited to a rotating (001) MgO single-crystal substrate (Supplement of Ref. [19]). The entire synthesis process was carried out under ultrahigh vacuum (5 × 10^−10^ mbar) in the molecular beam epitaxy (MBE) chamber manufactured by SPECS GmbH (SPECS, Berlin, Germany). The iron concentration profile in the palladium matrix was realized by a controllable variation of the iron evaporation cell temperature with a fixed palladium cell temperature. The temperature variation cyclogram was loaded in the Eurotherm 3504 temperature controller of the iron effusion cell. Immediately after synthesis, all films were high-temperature annealed at 873 K under ultrahigh vacuum conditions for their structural optimization. A detailed description of the stages of film synthesis and methods for controlling their structural perfection and chemical composition, as well as their thickness, were described in detail in our previous works [20,21,22]. The magnetic characteristics of the films were studied by vibration sample magnetometry (VSM) with the Quantum Design (San Diego, California, USA) PPMS-9 system and by magnetic resonance with the Bruker ESP300 continuous wave X-band spectrometer in a wide field range of 0–1.4 T and a temperature range of 20–300 K [19,21,22].

### 2.2. Simulations of Thermomagnetic Curves and Spin Waves

In Pd_1−*x*_Fe*_x_* alloys, the Curie temperature T_C_ and the saturation magnetization $M_{s}$ depend on the iron concentration [22]. Therefore, in Pd–Fe films with inhomogeneous iron distribution across the thickness, each sufficiently thin layer has its local $T_{C}(z)$ and $M_{s}(z)$. The profile of iron concentration in the film will manifest itself in the experimental dependence of the integral saturation magnetization on temperature $\bar{M_{s}}(T)$. Then the simulation of $\bar{M_{s}}(T)$ can be used as an indicator of whether the real iron concentration profile corresponds to its target shape. In the calculation of the dependence of saturation magnetization on temperature, we used the earlier established fact that the dependence $M_{s}(T/T_{C})/M_{s}(0)$ for all homogeneous epitaxial Pd_1−*x*_Fe*_x_* films (0 < *x* < 0.08) can be described by a general expression [23]:(1)$$\sigma (T,T_{C})=\frac{M_{s}(T)}{M_{s}(0)}={\left(\frac{\sqrt{\kappa^{2}+4(1-\kappa )(1-{\left(T/T_{C}\right)}^{3})/(1+p{\left(T/T_{C}\right)}^{3/2})}-\kappa }{2(1-\kappa )}\right)}^{0.5},$$
with κ=1.8 and p=− 0.02, deduced from the fit of the experimental data for Pd_1−*x*_Fe*_x_* [22]. Based on the predefined dependence of the iron concentration on thickness $c_{\mathrm{Fe}}(z)$, as well as the concentration dependence of the saturation magnetization $M_{s}(c_{\mathrm{Fe}})$ and the Curie temperature $T_{C}(c_{\mathrm{Fe}})$, and introducing $M_{s}(0,z)=M_{s}(T=0K,c_{\mathrm{Fe}}(z))$, $\sigma (T,z)=\sigma (T,T_{C}(c_{\mathrm{Fe}}(z)))$, the $\bar{M_{s}}(T)$ dependence for each sample can be calculated using the expression:(2)$$\bar{M_{s}}(T)=\frac{1}{d}\int_{0}^{d}M_{s}(0,z)\;\sigma (T,z)\;\;dz.$$

Figure 1a shows the dependences $M_{s}(c_{\mathrm{Fe}})$ and $T_{C}(c_{\mathrm{Fe}})$ used in the work in comparison with the experimental values for thin homogeneous films [22].

The linear approximation of $T_{C}\left(c_{\mathrm{Fe}}\left[\mathrm{at}.\%\right]\right)=52+19c_{\mathrm{Fe}}\;\left[K\right]$ was adopted for $c_{\mathrm{Fe}}\geq 2\;\mathrm{at}.\%$ in $\bar{M_{s}}(T)$ calculations (Figure 1a). A non-linear dependence of the saturation magnetization on iron concentration $\mu_{0}M_{s}(c_{\mathrm{Fe}}\left[at.\%\right])=0.023+0.058c_{\mathrm{Fe}}-1.8\cdot 10^{-3}{\left(c_{\mathrm{Fe}}\right)}^{2}\;[T]$ was used (Figure 1a). This dependence has also been used in all further calculations along with $\bar{M_{s}}(T)$.

In Figure 1b, a sketch is shown for a simulation of a standing spin wave (SSW) in a film. It is supposed that the magnetic properties of a film are varied along the normal to a film, and the magnetic field is applied in the same direction that coincides with the *z*-axis of the laboratory frame. An SSW in a ferromagnetic film with an inhomogeneous magnetization across the thickness is described by a linearized Landau–Lifshitz–Gilbert equation of motion for a resonant circular projection of magnetization $m\left(z,t\right)$ [15,19,24]:(3)$$\left[-D(z)\frac{\partial^{2}}{\partial z^{2}}+V(z)\right]m_{n}(z)=-H_{n}^{r}m_{n}(z).$$

Here, $D(z)=2A(z)/\mu_{0}M_{s}(z)$ is the normalized exchange stiffness coefficient, and
(4)$$V(z)=-\frac{2\pi f_{r}}{\gamma }-\beta (z)M_{s}(z)+\frac{D(z)}{M_{s}(z)}\frac{\partial^{2}M_{s}(z)}{\partial z^{2}},$$
where $f_{r}$ is the excitation frequency, γ is the gyromagnetic ratio and β(z) is the effective-to-saturation magnetization ratio, which is larger than 1 for easy-plane ferromagnets, while it is smaller than 1 for the easy out-of-plane ferromagnets.

Equation (3) has a structure of the Schrödinger equation with the boundary conditions of
(5)$$\frac{dm}{dz}+\alpha_{s}m=0,$$
where $\alpha_{s}=K_{s}/A_{s}$, $\alpha_{s}$ being the surface pinning coefficient, $K_{s}$ is the surface anisotropy energy and $A_{s}$ is the exchange stiffness constant at the surface. In the case of $\alpha_{s}=0$, the spin boundary conditions are free, while in the case of $1/\alpha_{s}=0$, the spin boundary conditions are fixed, or pinned. By the analogy with the Schrödinger equation, *V*(*z*) plays a role of potential energy which has a shape of a potential well. Within such an approach, $\left(-H_{n}^{r}\right)$ is the *n*-th “energy level” of the standing spin waves in the potential well *V*(*z*) given by Equation (4). Therefore, in this work, the visualization of the SSW modes is a filling of the potential well by the waves at ${\left(-H_{i}^{r}\right)}_{n}$ levels, as it was done in [15,25]. The calculation was performed assuming $\alpha_{s}=\mathrm{const}$, D(z)=const and β(z)=const, which are the same for all samples. For certain $M_{s}(z)$ profiles, such as a linear one, it is quite a good approximation, but for the others it allows to catch basic qualitative features of the SSW.

An intensity of the resonant line in the spectrum was calculated following the expression:(6)$$I_{n}\propto \int h_{rf}(z)\;m_{n}(z)M_{s}(z)\;dz\propto \int \cosh\left[(z-d/2)/\sigma_{f}\right]m_{n}(z)M_{s}(z)dz,$$
where $h_{rf}$ is the distribution of the rf magnetic field within the film and $\sigma_{f}$ is the ferromagnetic skin depth [19,26]. The resonance field and line intensities calculated from Equations (3)–(6) were used to construct a spin wave resonance spectrum. The linewidth estimation from the frequency dependence of the first SSW mode in Ref. [19] has shown that, at the measurement frequency of ~9.44 GHz, the contribution of the inhomogeneous line broadening and the intrinsic Gilbert damping are almost equal, and the Gilbert damping parameter *α* ≈ 0.007 (because of possible concentration dependence, the estimated value of *α* should be considered as the “effective” one). For the actual calculations, the integral linewidth taken from the experimental data and the Lorentzian line shape were used. For more details on the SSW spectrum calculation, see [19].

## 3. Results and Discussion

### 3.1. Temperature Dependences

In Figure 2a–e, the measured $\bar{M_{s}}(T)$ dependences for all twelve samples of Table 1 are shown (symbols) and compared with the calculated $\bar{M_{s}}(T)$ dependences (line) according to Equation (2). One can see that the $\bar{M_{s}}(T)$ dependences of Lin50–400 samples practically coincide, which confirms the unified character of the iron concentration profile in these films (Figure 2a). The same is evident for Sin200 and Cos200 films (Figure 2e). In films with a stepped profile of magnetic properties, the kink is clearly visible in the thick BiL200 structure and becomes blurred as the thickness of the film decreases (Figure 2b). In general, the experimental dependences for all films are reasonably well described by Equation (2) with the profiles $c_{Fe}(z)$ defined by the deposition procedure (profiles are shown in the insets to the panels).

The temperature evolution of the magnetic resonance spectra of sample Lin200 is shown in Figure 2f. The decrease in the resonance field values $H_{n}^{r}(T)$ with increasing temperature is associated with a gradual decrease in both the magnetization and the effective thickness of the ferromagnetic layer, the latter because of $T_{C}(z)$ dependence. This trend is the typical feature for all investigated samples. All of the resonance spectra studied further were obtained at 20 K, close to the low-temperature plateau of the $\bar{M_{s}}(T)$ dependences of the samples.

### 3.2. SSW Resonance Spectra of Linear-Profile Samples Lin50–400

Figure 3 shows the spin wave resonance spectra for Lin50–400 samples. As one can see, the number of modes increases with the film thickness, which is typical for standing spin waves in thin films. A set of the collected spectra was used to adjust the parameter values $\alpha_{s}=-\;0.05$, D=16 T·nm^2^, $\sigma_{f}=170$ nm and $\beta_{k}=1.35$ that have not been varied for the rest of the samples. The value of $\beta_{k}>1$ indicates the hard direction along the normal to the film, which is verified by the shape of the magnetization curves in the out-of-plane geometry (not shown here, see Ref. [19]). A small value of $\alpha_{s}$ testifies insignificant pinning of the spins at the film surface. The inaccuracy of the *x*(*z*) profile and the film thickness measurement could be at the origin of the *n* = 3 mode discrepancy in Figure 3a.

Consideration of the spatial distribution of the magnetization precession amplitude *m*(*z*) for each SSW mode in its resonant field showed that, at the potential well *V*(*z*) boundary, the amplitude drops to zero, i.e., the dynamic pinning occurs [15]. Most of the absorption also occurs near the potential boundary.

### 3.3. SSW in Bilayer Samples BiL20–200

Studies of bilayer samples are intriguing in terms of the formation of spin waves in a structure with extended homogeneous regions though inhomogeneous as a whole. In this case, the model no longer describes the experimental data well, although its predictions qualitatively fit the experimental observations (Figure 4).

In the thin BiL20 sample, which does not accommodate a single standing wave, only one resonance line is observed (Figure 4a). The simulation indicates (Figure 4b) that the precession amplitude does not vary significantly over the thickness (which is close to a classic FMR). The resonance field corresponds approximately to a mean value of the two-layer potential. As the thickness of the bilayer increases, two modes become observable (Figure 4c,d), similar in character to the acoustic (*n* = 1) and optical (*n* = 2) coupled precession modes. With the further increase in thickness (Figure 4e,f), the second intense resonance line appears in the spectrum; here, most of the absorption occurs in the layer with lower magnetization. In a sufficiently thick bilayer (Figure 4g,h), this trend is manifested even brighter: in addition to two quasi-FMR intense modes (*n* = 1 and 5), originating from the layers with higher/lower magnetization, respectively, at their local potentials, several much less intense modes are resolved.

Magnetic resonances in bilayer systems were studied extensively during the last decades, both experimentally and theoretically [27,28,29,30,31,32,33,34,35,36,37,38]. In some of the works, the coupled precession with the acoustic and optical modes formation was considered ([27,28,29,30,31,32] and references therein). The others dealt with spin wave resonances for particular realizations of the layered magnetic structure [33,34,35,36,37,38]. The theoretical approach similar to ours is presented in Ref. [39], though without comparison with an experiment. An important result of our work is an observation of multiple SSW modes in bilayer samples. This fact, as well as the successful semi-quantitative reproduction of the SSW spectra for all four samples within a unified spin-wave excitation approach, in our opinion, indicates its improved reliability and applicability to a wider range of complex magnetic systems and phenomena.

### 3.4. SSW in the Trilayer Structure and Films with Non-Linear Magnetization Profiles

The TriL220 and Lor400 samples were fabricated to demonstrate how, by varying the magnetization profile across the thickness of the film, one could dramatically modify the character of the SSW spectrum. For this purpose, suitable profiles were initially calculated within the model of Equation (3). Then the samples with the specified profiles were deposited. In the TriL220 sample, additional effective pinning at the interfaces was created by sandwiching the thick homogeneous layer with thin layers of significantly lower magnetization. According to the preliminary simulation, this should result in the SSW spectrum with $(H_{1}^{r}-H_{n}^{r})\propto n^{1.92}$. In the Lor400 sample, the profile was calculated so that the resonances follow the dependence $(H_{1}^{r}-H_{n}^{r})\propto n^{0.58}$. Figure 5 approves the qualitative agreement of the model predictions with our experiment. However, a notable disagreement takes place for the TriL220 sample with the stepped character of the profile, since, as it was shown above, the model does not describe the systems with sharp magnetization drops as well.

We note also that antisymmetric modes—the modes with an even index and zero intensity—should exist in samples with a symmetric magnetization profile [15]. Such modes have indeed been identified and are shown by dashed violet lines in Figure 5b,d. In the experiment, such low-intensity modes (*n* = 2 and 4) were observed in the Lor400 sample.

The Sin200 and Cos200 samples are integrally identical in terms of thickness, concentration drops and drop characters. The integral similarity of the samples is manifested in an almost identical temperature dependence of $\bar{M_{s}}(T)$ (see Figure 2e). However, the SSW analysis within the model Equation (3) predicted a significant difference in the magnetic resonance spectra of these samples (preliminary modeling and its results were discussed in our recent paper [19]). This dissimilarity originates from the difference in potential wells *V*(z) in these samples, that, consequently, are filled differently by the SSW modes (Figure 6b,d). Our experiments have confirmed the predictions of Ref. [19].

According to the simulations [19], in the Cos200 sample, the symmetric (odd *n*) and antisymmetric modes (even *n*) should have the same resonance fields. In the experiment, however, the even and odd modes split and have comparable intensities. In our opinion, this indicates some asymmetry of the actual potential wells, which lifts the degeneracy and violates the condition of the antisymmetric modes occurrence (Figure 6d). Indeed, an introduction of a small asymmetry to the concentration profile (of ~0.5 at.%) in the simulation allowed us to reproduce the splitting pattern and the line intensities in the experimental spectrum (Figure 6c). The origin of the weak lines 2′ and 4′ in Figure 6c with the properties characteristic of antisymmetric modes remains unclear.

To visualize the drastic differences between the SSW resonance spectra of graded ferromagnetic films and the “classic” $n^{2}$ dependence for homogeneous films, we compared the latter with the fits of experimental SSW resonance field patterns of the SSW with the power law $(H_{1}^{r}-H_{n}^{r})\propto n^{\delta }$ (Figure 7). The results clearly demonstrate a possibility to manipulate the standing spin wave spectrum in thin film structures by varying the magnetization profile. In contrast to “classic” power “2” for the homogeneous ferromagnetic films, the fitting parameter *δ* changes within a broad range from 0.27 (for Lor400 sample) to 2.28 (for TriL220 sample). In single-well potential structures, the shape of the potential well determines the character of the SSW spectrum. At the same time, for more complex profiles with several spatially separated potential wells, the dependence has a more complex character determined, in addition to the shape of the wells, by the order of their filling. This is clearly demonstrated in samples Sin200 and Cos200 (Figure 7b).

Based on the analysis of the SSW data collected for twelve epitaxial Pd–Fe films with different iron concentration profiles, we conclude that the model Equations (3)–(6) used for calculating the SSW spectra describe the experimental results well. However, this model is better applicable to films with smooth profiles rather to sharp/stepped ones. The possible reasons of discrepancy between experimental and theoretical spectra include the oversimplified definition of the normalized exchange stiffness coefficient *D*(*z*) in Equation (3), as well as the possible redistribution of Fe and local mechanical stress at internal interfaces of films with a sharp gradient of Fe content.

## 4. Conclusions

Magnetic resonances due to standing spin waves in Pd–Fe films with different distributions of the magnetic properties across the thickness have been studied. Variation of the profile shape allows us to predictably modify the SSW resonance spectrum, e.g., the dependence of the resonance field on the mode index within our samples set varied from $(H_{1}^{r}-H_{n}^{r})\propto n^{0.27}$ to $(H_{1}^{r}-H_{n}^{r})\propto n^{2.28}$. Analysis of the SSW spectra provided the following common parameters for the ferromagnetic Pd–Fe films: the normalized exchange stiffness constant D=16 T·nm^2^, the surface pinning coefficient $\alpha_{s}=-\;0.05$, the skin layer depth $\sigma_{f}=170$ nm and the ratio of the effective magnetization to saturation magnetization $M_{eff}/M_{s}=\beta_{}=1.35$. The obtained results indicate that the approach suggested and realized in this work can be applied to engineer the spin waves dispersion in graded ferromagnetic films.

## Figures and Tables

**Figure 1 nanomaterials-12-04361-f001:**
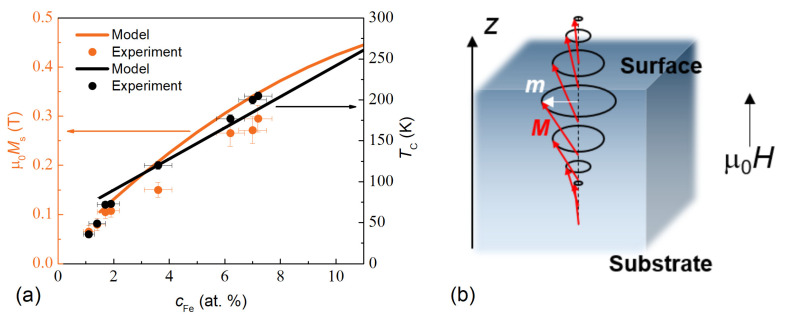
(**a**) Dependences of the saturation magnetization $M_{s}(c_{\mathrm{Fe}})$ (left axis) and the Curie temperature (right axis) on iron concentration $c_{\mathrm{Fe}}$ in Pd–Fe films used in the calculations (lines) and corresponding experimental data for homogeneous thin films (symbols). (**b**) A sketch for a simulation of the standing spin waves in a film.

**Figure 2 nanomaterials-12-04361-f002:**
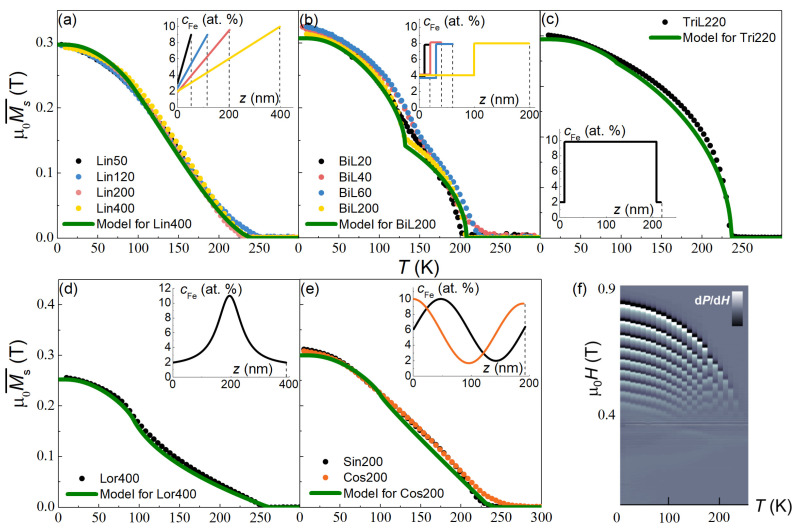
Measured $\bar{M_{s}}(T)$ dependences (symbols) of the samples listed in Table 1 with the magnetic field of μ_0_*H* = 5 mT applied along the easy direction in the film plane (**a**–**e**). Lines—the calculated $\bar{M_{s}}(T)$ dependences following Equation (2); in the insets to panels (**a**–**e**), the iron concentration profiles are presented. (**f**)—temperature evolution of the magnetic resonance spectra of sample Lin200 in the out-of-plane arrangement.

**Figure 3 nanomaterials-12-04361-f003:**
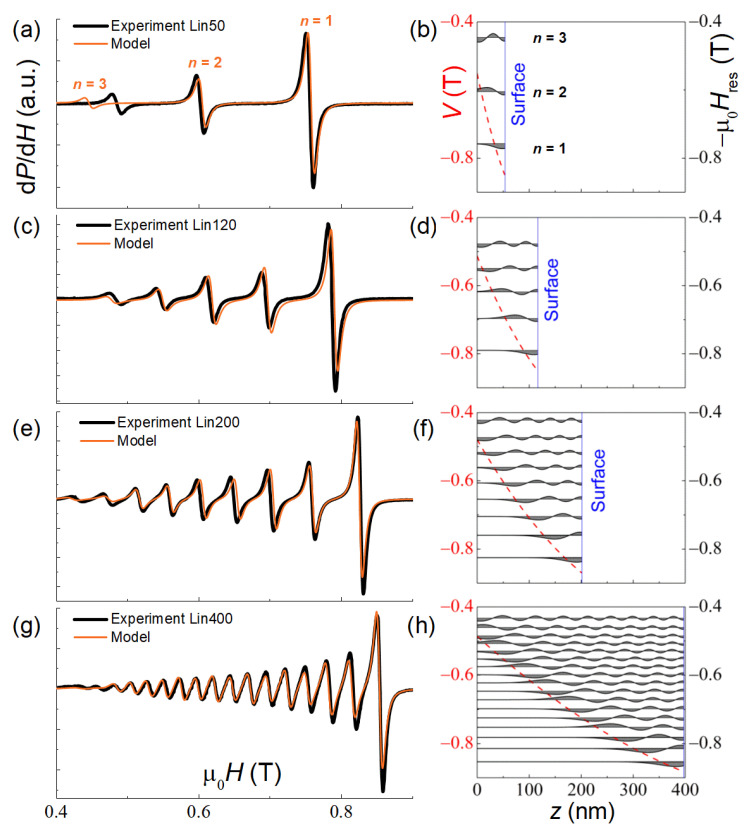
Measured (black lines) at *T* = 20 K and simulated (orange lines) SSW resonance spectra of the Lin50–400 samples (**a**,**c**,**e**,**g**). In the right-hand panels, the spatial distributions of the magnetization precession amplitude $m_{n}(z)$ are shown for each mode in its resonance field $\left(-H_{n}^{r}\right)$ marked by the horizontal lines (**b**,**d**,**f**,**h**). The red dashed line is the potential well profile *V*(*z*).

**Figure 4 nanomaterials-12-04361-f004:**
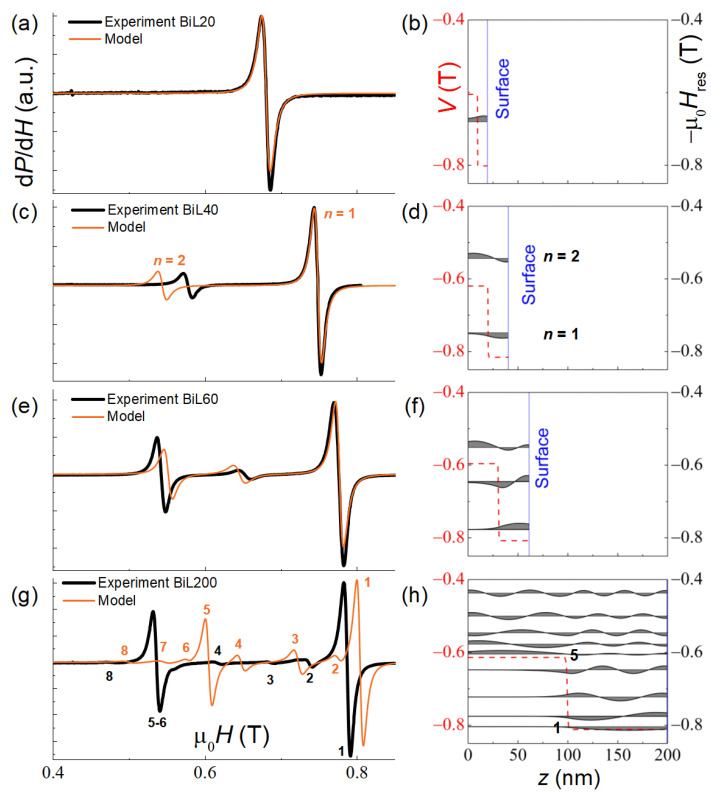
Experimental (black lines) and simulated (orange lines) spectra of SSW resonances at *T* = 20 K of the bilayer BiL20–200 samples (**a**,**c**,**e**,**g**). In the right-hand panels, the spatial distribution of the magnetization precession amplitude $m_{n}(z)$ for each mode is displayed (**b**,**d**,**f**,**h**).

**Figure 5 nanomaterials-12-04361-f005:**
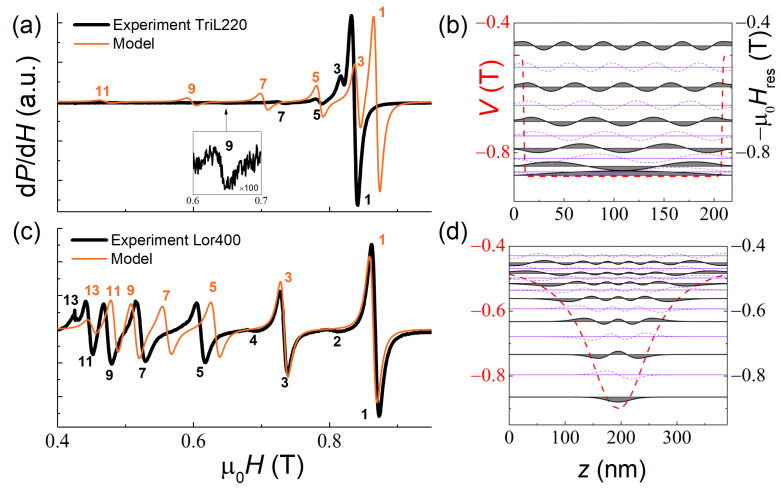
Experimental (black lines) and simulated (orange lines) SSW resonance spectra of the TriL220 (**a**) and Lor400 (**c**) samples at *T* = 20 K. In the right-hand panels, the spatial distribution of the magnetization precession amplitude $m_{n}(z)$ for each mode is shown (**b**,**d**); the dashed violet lines indicate the antisymmetric modes.

**Figure 6 nanomaterials-12-04361-f006:**
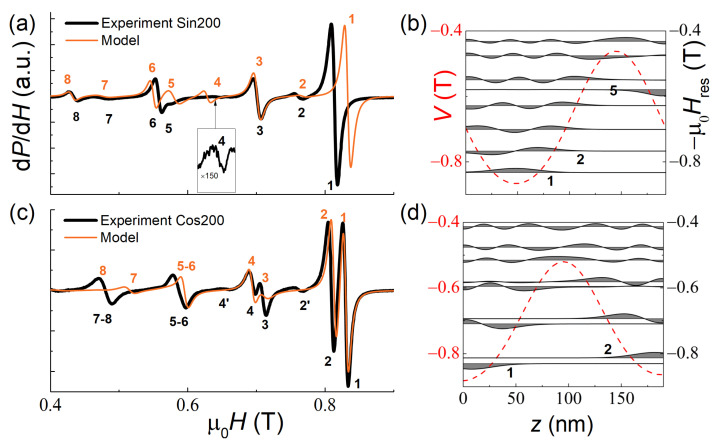
Experimental (black lines) and simulated (orange lines) SSW resonance spectra of the Sin200 (**a**) and Cos200 (**c**) samples at *T* = 20 K. In the right-hand panels, the spatial distribution of the magnetization precession amplitude $m_{n}(z)$ for each mode is shown (**b**,**d**).

**Figure 7 nanomaterials-12-04361-f007:**
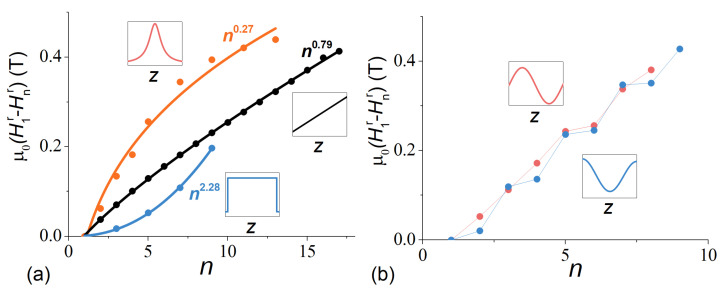
Experimental resonance field difference between the 1st and n-th SSWs as a function of the mode index for TriL220, Lin400 and Lor400 (**a**), and for Sin200 and Cos200 (**b**), samples. The lines present the best fit results with $(H_{1}^{r}-H_{n}^{r})\propto n^{p}$ function (**a**).

**Table 1 nanomaterials-12-04361-t001:** Samples description.

Profile Type	Label	Thickness, nm	*c* _1_	*c* _2_
Linear	Lin50	53	3 ^a^	9 ^b^
	Lin120	116	2.4 ^a^	9 ^b^
	Lin200	202	2 ^a^	9.6 ^b^
	Lin400	397	2 ^a^	10 ^b^
Bilayer	BiL20	10/10	3.9	7.8
	BiL40	20/20	4.1	8.1
	BiL60	30/30	3.7	7.9
	BiL200	100/100	4	8
Trilayer	TriL220	10/200/10	2 ^c^	9.8
Lorentzian	Lor400	390/100 ^d^	2 ^a^	11 ^b^
Sine	Sin200	193 ^e^	2 ^a^	10 ^b^
Cosine	Cos200	190 ^e^	2 ^a^	10 ^b^

^a^ Minimal concentration. ^b^ Maximal concentration. ^c^ Concentration in the first and third layers. ^d^ Width at half-maximum. ^e^ The period of Sin and Cos fits the thickness of the film.

## Data Availability

The data presented in this study are available on request from the corresponding author.

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
