# Peer review of "Engineering the Exchange Spin Waves in Graded Thin Ferromagnetic Films"

_nanomaterials, 2022, doi:10.3390/nano12244361_

Round 1

Reviewer 1 Report

The manuscript by Yanilkin et al. presents a study on standing spin waves formed in ferromagnetic thin films of PdFe alloy with various yet controlled iron compositions across the thickness.  Standing spin wave spectra and its temperature dependence in different samples were measured experimentally. Theoretical modellings yield results quantitatively or qualitatively agreeing with the experimental results. This work demonstrates a possible approach for spin wave engineering by modifying the magnetic properties of magnetic thin films across the thickness. In my opinion, it should be useful for the development of novel magnonic devices based on the manipulation of spin waves in magnetic nanostructures. Before I can recommend its publication, there are a couple of issues I would like to ask the author to address.

To utilize spin waves to carry information, it usually requires the magnetic materials to have small damping coefficient and thus slow attenuations of spin waves propagating inside. How about the damping property of PdFe alloy studied in this work? Dose it depend on the iron composition?  

It should be helpful for the audience to provide a short introduction of the theoretical model used in the calculations. For the samples studied, is it practicable to carry on micromagnetic simulations to study the magnetization dynamics?

Author Response

We gratefully acknowledge remarks and suggestions of Reviewers. Below, please, find our responses to the remarks and suggestions. Amendments made in the manuscript are given after the answers and are indicated by the yellow background.

We sincerely thank all three Reviewers whose remarks and suggestions helped to improve the presentation quality.

Reviewer 2 Report

This is an interesting technical manuscript on a modern topic of spin-wave physics in ferromagnets. Although I do not have strong objections against its publication, it needs improvement before possible acceptance. My suggestions are presented below. 1. Physics behind the results. The authors claim observation of dependencies on the mode index as (H_{1}^{r} − H_{n}^{r}) \sim n^{0.27} and n^{2.28}. However, absolutely no physical qualitative explanations of these powers has been given. These n-dependencies have to be clearly explained in the paper. A reference to published theoretical or numerical papers will be insufficient. 2. The authors are missing a connection to modern research topic of spin torques
as a tool for exciting spin waves. I just mention some papers on the topic: Nature 511, 449 (2014), Phys. Rev. B 96, 014408 (2017), Phys. Rev. Lett. 119, 077702 (2017), J. Phys. D: Appl. Phys. 50, 265004 (2017), Phys. Rev. B 97, 134402 (2018), Nature Materials 17, 800 (2018), APL Materials 9, 060901 (2021) and suggest the authors to discuss the effect of spin torque on the spin waves excitation.
3. Presentation in the text. 3.1 Line 115. It is better to use “/” for the derivative of m over z, not the direct fraction.
3.2. Line 120, long in-line equation for V(z) has to be made a separated one and numbered.

Author Response

(The authors gave the same response as above.)

Reviewer 3 Report

The article presents an interesting study of standing spin waves in different magnetic structures, specially varying the content of Fe. The article deserves publication as long as some minor changes are introduced.

-Page 3 line 109: “SSW in an inhomogeneous across ..” ¿homogeneous what?

-I recommend a more clear presentation of the approach shown in equation 1. I would mention that Equation 1 is general and includes the reference 21. And then for this system the authors have derived the factor p and kappa and then cite reference 22. This can be done easily by modifying maybe a sentence.

-The models in Figure 1 are not well described. What model do they use for the Curie temperature? It seems they use a linear approximation with the Fe content. If that is the line, then why the crossing is not at zero K because I suppose that without Fe, Pd is not magnetic. Also the authors include later the nonlinear fit to the model in page 5 line 168. I would include it already here.

-The authors should define V just after equation 3. The way it is introduced now is a bit complicated. It is mentioned that there is a potential and that then the solution is the one when the potential has this form. I think it is a bit better to first define V.

-Beta is not defined immediately after line 120. It is related to the effective anisotropy but it does not explicitly mention what is the relation and what is the configuration of the anisotropy. I think at least a hint about the anisotropy configuration should be included in the text. I understand it may be also only shape anisotropy but anyway this fact should be mentioned, if that if the case.

-The authors consider D and beta constants but by configurations they depend on z through Ms at least. I think this is the weakest point of the article. The authors return to this point in lines 265 and 266. I think it would be better to address this at least giving a justification for this simplification that seems to work in some samples. Ideally, they should do the calculations with D and beta dependent on z, but maybe this is out of the scope of this article.

-Line 135. I think the authors meant Figure 2 a-e

-In Figure 3.a The mode 3 is not well represented by the model. It is a bit strange that all the other modes in the whole figure are very accurately represented but not this one. This is not discussed in the text. Could the authors argue in the manuscript about the possible origin of this discrepancy?

-The authors use p for one parameter in equation 1 and for the exponent in line 249. I recommend using a different letter.

-Line 178-179: “The most of the absorption also occurs near the potential boundary.” should be “Most of the absorption …”

-The FMR and the modes in a bilayer have been discussed previously in the literature. At least in the discussion a comparison with previous results would improve it.

Author Response

(The authors gave the same response as above.)
